# Efficacy of Nasal High-Flow Oxygen Therapy in Chronic Obstructive Pulmonary Disease Patients in Long-Term Oxygen and Nocturnal Non-Invasive Ventilation during Exercise Training

**DOI:** 10.3390/healthcare10102001

**Published:** 2022-10-11

**Authors:** Valeria Volpi, Eleonora Volpato, Elena Compalati, Marius Lebret, Giuseppe Russo, Salvatore Sciurello, Gabriele Pappacoda, Antonello Nicolini, Paolo Banfi

**Affiliations:** 1IRCCS Fondazione Don Carlo Gnocchi, 20148 Milan, Italy; 2Department of Psychology, Università Cattolica del Sacro Cuore, 20123 Milan, Italy; 3MedLab, Air Liquide Medical Systems, ERPHAN, Paris-Saclay University, 92160 Antony, France

**Keywords:** chronic obstructive pulmonary disease, humidified high-flow nasal cannula, humidified high-flow nasal therapy, high-flow oxygen therapy, pulmonary rehabilitation, exercise tolerance

## Abstract

High-flow oxygen therapy (HFOT) improves gas exchange and dead space washout and reduces the level of work required for breathing. This study aimed to evaluate pulmonary rehabilitation (PR) combined with HFOT in COPD patients treated with nocturnal non-invasive ventilation (NIV) and long-term oxygen therapy (LTOT). In particular, we sought to discover whether the addition of HFOT during exercise training could improve patients’ performance, mainly with regard to their Six-Minute Walking Test (6MWT) outcomes, and reduce the exacerbation rates, periods of rehospitalization or need to resort to unscheduled visits. Thirty-one COPD subjects (13 female) who used nocturnal NIV were included in a randomized controlled trial and allocated to one of two groups: the experimental group (EG), with 15 subjects, subjected to PR with HFOT; and the control group (CG), with 16 subjects, subjected to PR without HFOT. The primary outcome of the study was the observation of changes in the 6MWT. The secondary outcome of the study was related to the rate of exacerbation and hospitalization. Data were collected at baseline and after one, two and three cycles of cycle-ergometer exercise training performed in 20 supervised sessions of 40 min thrice per week, with a washout period of 3 months between each rehabilitation cycle. Statistical significance was not found for the 6MWT distance (W = 0.974; *p* = 0.672) at the last follow-up, but statistical significance was found for the Borg scale in regard to dyspnea (W = 2.50; *p* < 0.001) and fatigue (W = 2.00; *p* < 0.001). HFOT may offer a positive option for dyspnea-affected COPD patients in the context of LTOT and nocturnal NIV.

## 1. Introduction

The course of chronic obstructive pulmonary disease (COPD) is characterized by periods of exacerbation, leading to the worsening of the clinical state of patients and requiring hospitalization, additional visits and, often, ventilatory support [1]. The treatment of patients with COPD and acute respiratory failure with non-invasive ventilation (NIV) improves outcomes [1,2], but persistent hypercapnia, associated with increased mortality [3,4] and early re-hospitalization, might persist [5,6].

Murphy et al., 2017 [7] showed that among patients with persistent hypercapnia, following an acute COPD exacerbation, the combination of home NIV with home oxygen therapy prolonged the time to re-hospitalization or death by 12 months.

Moreover, several studies on stable COPD patients with or without chronic hypercapnic respiratory failure have shown that nasal high-flow oxygen therapy (HFOT) reduces the respiratory rate [8,9,10], dead space [11] and work required for breathing [12,13] and improves gas exchange [8,9,10,14]. Other studies have shown that there are no significant differences in the endurance time during exercise between patients with and without HFOT who are recovering from an acute COPD exacerbation and patients in a stable condition. Other authors found that the addition of HFOT was not associated with an improvement in the endurance time; however, a greater improvement in the 6 min walking distance (6MWD) was observed in the HFOT group, although the *p*-value of the primary analysis did not reach the threshold of significance [15,16].

No studies, to our knowledge, have been conducted on hypercapnic COPD patients undergoing long-term oxygen therapy (LTOT) and non-invasive nocturnal ventilation (NIV) at home, in addition to respiratory rehabilitation (RR) cycles with HFOT. 

In our single-center randomized clinical trial, we hypothesized that cycles of respiratory rehabilitation with HFOT applied to a group of COPD patients undergoing long-term oxygen therapy and nocturnal non-invasive home ventilation could improve the patients’ performance, especially in the outcomes of the 6MWD, and reduce the exacerbation rate and/or periods of re-hospitalization or the need to resort to unscheduled visits, compared to similar patients who underwent only cycles of respiratory rehabilitation with oxygen therapy.

## 2. Materials and Methods

### 2.1. Study Design and Data Collection

This was a prospective randomized controlled trial involving a single center, the IRCCS Santa Maria Nascente, Fondazione Don Carlo Gnocchi (Milan), and it included subjects affected by COPD undergoing LTOT and nocturnal non-invasive ventilation (NIV) treatment. The study protocol was defined according to the Consolidated Standard of Reporting Trials (CONSORT) guidelines and was approved by the Ethics Committee of IRCCS Fondazione Don Carlo Gnocchi, IRCCS Regione Lombardia (11-12-2019). Written informed consent was obtained from all the patients before they entered the study. The study was conducted following the principles of the Declaration of Helsinki. The study was also registered at Clinical.Trials.gov, ID: NCT04683952. 

### 2.2. Participants

The recruitment of the participants was carried out from January 2020 to October 2021, with an interruption due to COVID-19 lockdowns from February to September 2020, at the IRCCS Fondazione Don Gnocchi in Milan (Italy). 

Participants were included if they fulfilled the following criteria:A diagnosis of moderate to severe chronic obstructive pulmonary disease (COPD) according to Vogelmeier et al. [17].Age ranging from 18 to 80 years.Nocturnal non-invasive-ventilation (NIV) prescription according to the ATS/ERS guidelines, with long-term oxygen therapy [18].Clinical stability (no exacerbation and no change in, or addition of, respiratory drugs in the last month) [19].No COVID-19-related infections associated

The exclusion criteria were: Orthopedic or neurological pathologies that limit the patient’s physical performance.Cognitive impairment (Mini-Mental State Examination < 24) [20].Advanced heart disease (NYHA class > 2) classes of heart failure (American Heart Association) [21].NIV compliance that is inferior to 5 h per night [22].

### 2.3. Interventions

After screening, the patients were randomized (1:1) using a dedicated software program (E.C. https://www.randomizer.org/, accessed on 8 January 2020) and assigned to either the experimental group (O_2_ with HFOT) or the control group (only O_2_). This was a single-blind study, and only the statistician involved in the analysis was blind to the group allocation. 

Participants were allocated to one of two study groups:

(1) The experimental group (EG): LTOT COPD patients undergoing nocturnal NIV (VEMO 150 EOVE^®^ Vitalaire Italia, Assago-Milan, Italy) who were subjected to the protocol for respiratory rehabilitation with HFOT according to a medical prescription. Patients performed training sessions with the concurrent administration of HFOT through the VEMO 150 EOVE^®^ device (Appendix A). This system generates flows of up to 60 L/min of humidified, heated air (from 30 to 37 °C) by altering the FiO_2_ within the system itself. Air is administered through an open circuit using an Optiflow™ nasal cannula (Fisher&Paykel, Auckland, New Zealand), delivering flows directly into the nares. The airflow was set to 60 L/min and the temperature was set to 37 °C, according to the patient’s level of tolerance. Every modification was recorded. FiO_2_ was set according to the run-in phase.

(2) The control group (CG): LTOT-COPD patients undergoing nocturnal NIV who were subject to the same protocol for respiratory rehabilitation with oxygen alone, without HFOT application, according to a medical prescription. Participants in the control group performed sessions with oxygen administered through a Venturi mask (Appendix A).

### 2.4. Run-In Phase

After the assessment used to establish the workload, each patient performed a training session using the cycle ergometer. This process was performed through a six-minute walk test, whose distance was then inserted into the formula that was used for the definition of the workload. The intensity of the workload established for this study was 60–80% of the Wmax calculated using Hill’s formula:Wmax = (0.122 × 6MWD) + (72.683 × Height) − 117.109

The amount of oxygen administered during training was set at the beginning of the program: EG and CG patients underwent a 15 min run-in phase on a cycle ergometer with a FiO_2_ and were able to maintain a pulse oximetry of >92% throughout the session.

### 2.5. Exercise Training Program

The cycle ergometer exercise training was performed by all the participants. It consisted of 20 supervised sessions of 40 min thrice per week, and every session was divided into three phases:Warm-up: 5 min at 0 watts.Training: 30 min of resistance training at 60–80% of the maximal workload. Patients had to maintain a cycling rate ranging between 40 and 50 rpm.Cooldown: 5 min at 0 watts.

Variations in intensity by 10 watts were applied, according to Maltais et al. [23]. The workload was increased when patients showed dyspnea and/or leg fatigue with a score of less than 4 on a modified CR-10 Borg scale. The workload was reduced or unchanged if the Borg score [24] was equal to or above 4 or 5.

During training and the run-in phase, the physiotherapist took note of the FiO_2_ and the dyspnea at the beginning and end of the session using the Borg RPE and again when the patient performed the 6MWD.

### 2.6. Measurements

At baseline (T0), demographic data were collected, along with other clinical data, such as spirometry, a blood gas analysis, the modified Medical Research Council scale (mMRC), dyspnea perception (Borg modified scale), 6MWD, COPD assessment test (CAT) and Saint George Respiratory Questionnaire (SGRQ) about health-related quality of life (HRQL). In the table below (Table 1), the tools and measures adopted for about four assessment times are indicated:4.T0: baseline.5.T1: at the end of the first 20 sessions of the rehabilitation cycle.6.T2: at the end of the second 20 sessions of the rehabilitation cycle.7.T3: at the end of the third 20 sessions of the rehabilitation cycle.

The six-minute walk test was performed following the latest guidelines [24,25], using a 30 m-length hallway with two cones limiting the circuit. A respiratory therapist supervises the entire test with a chronometer, a rev counter, a modified Borg scale [26], a sphygmomanometer (for the arterial pressure) and an oximeter (NONIN Medical Inc, Medicare, Origgio, Italy), recording all the necessary data during the test (cardiac frequency, oxygen saturation).

Blood gas analyses were performed through an arterial blood withdrawal of 2 mL of arterial blood from the radial artery using a specific needle and syringe and the use of a blood gas analyzer (GEM 4000 Premier Plus, Werfen, Instrumentation Laboratory).

CAT, mMRC and STGQ were performed by the patient, supervised by the respiratory therapist.

A lung functional test (spirometry) was performed by a respiratory therapist following the latest guidelines [27] and using a plethysmograph (Master Screen Body Jaeger Vyntus™ Pneumo, Vyaire, Mettawa, IL, USA).

### 2.7. Outcome Measures

The primary outcome measure was the change from baseline to three months in the 6MWD (minimal clinically important difference: 30 m for hospitalization [28] and 54–80 m regarding the efficacy of the pulmonary rehabilitation [29]).

The rates of exacerbation, unplanned visits to general practitioners, emergency department visits, hospitalization and admission to intensive care, as well as changes in the mMRC, CAT and SGRQ, were considered as secondary outcomes.

### 2.8. Statistical Analysis

For all the outcome measures, summary descriptive statistics were calculated at baseline to assess any changes in the scores from baseline to T1, T2 and T3 by study group and between times. Given the relatively small sample size and our expectation of a non-Gaussian distribution, we opted for the use of non-parametric tests. The differences between the two groups were then analyzed using the Mann–Whitney U-Test, while the internal analyses were conducted using the Wilcoxon test. The significance level was set at 0.05. All analyses were performed using the statistical software Jamovi (version 2.3.3).

## 3. Results

A pool of 523 moderate to very severe COPD individuals were identified. Among them, 462 were then excluded because NIV treatment was not prescribed. A total of 30 possible participants were excluded because they did not meet the inclusion criteria (Appendix A), and 31 COPD subjects finally took part in the study. All patients in the study sample were receiving inhalation therapy. Out of the total, 74% had triple therapy and the remaining percentage received double therapy. In particular, for the intervention group, 75% followed a triple therapy program, while in the control group, the same therapy was received by 73% of the patients. All the subjects participated until the end of the study. At baseline, the experimental and the control group were compatible regarding their sociodemographic features, the blood gas analysis, spirometry and other outcome measures (Table 2).

### 3.1. Primary Outcome

Over time, no statistical significance in the 6MWD at each follow-up was identified, both within and between groups. However, statistical significance was found for the Borg scale, specifically at T0-T1 (*p* < 0.001). At T0-T2, there was statistical significance only for the Borg scale (*p* < 0.001).

We compared the Borg scale values acquired between baseline and the three follow-ups. Statistical significance was reached only in the case of the first two follow-ups (T0-T1, *p* < 0.001; T0–T2, *p* < 0.001). Regarding the final follow-up, statistical significance was found for the final Borg dyspnea (*p* < 0.001) and final Borg fatigue (*p* < 0.001) scores, as shown in Table 3.

### 3.2. Secondary Outcomes

Our qualitative analysis showed a trend of reduction in the rate of exacerbations in both groups over time. In the experimental group, the exacerbation dropped from an average of 1.46 at the baseline to 0.66 at T3. In the control group, baseline exacerbations showed an average of 1.68, and at T3, the result was 0.18. Additionally, we identified a decrease in the rate of re-hospitalization in both groups, dropping from an average mean of 1.2 (T0) to 0.4 (T3) in the control group, while the experimental group showed a decrease from 1.75 (T0) to 0.37 (T3), but these results did not reach statistical significance. Conversely, in regard to the health status and dyspnea scores, only the mMRC scale reached statistical significance throughout the period between the baseline and the last follow-up (T0–T3 *p*: 0.033).

## 4. Discussion

The six-minute walk test is a reproducible field test that is often used as the primary outcome in a wide variety of clinical trials investigating the effects of the intervention of respiratory rehabilitation. As a primary measure (6MWD), it is a recognized important mortality predictor [30,31,32,33]. The 6MWD can also identify groups of COPD patients at higher risk of exacerbation-related hospital admission. Another outcome of the 6MWD is desaturation, which is associated with increased mortality, lung function decline and an increased number of exacerbations.

Murphy et al. found that a 1-year reduction in the 30 m distance walked during the test was associated with an increased risk of death over the subsequent 12 months. The same conclusion was not reached in regard to the hospitalization and exacerbation rates [7]. It is of paramount importance to highlight that, in the same study, the investigators enrolled COPD individuals from the post hoc ECLIPSE study, while in our study the sample selection was meticulously performed to investigate a specific COPD clinical phenotype among parents with a prescription for LTOT and nocturnal NIV.

Other studies have suggested that HFOT leads to improved 6MWD scores and arterial oxygen saturation, but none of the enrolled patients with hypercapnic respiratory failure were treated with home NIV [34].

In our single center, in a randomized clinical trial that enrolled COPD patients with hypercapnic respiratory failure who received treatment with OTLT and home NIV, we found that the distance walked by both groups did not reach a statistically significant variation, even though the same observation was clinically relevant both regarding the T0 results and results between groups. More specifically, as reported in the literature, the 6MWD score should change by approximately 35 m for patients with moderate to severe COPD in order to represent an important effect [35].

Other studies examined the application of high-flow oxygen during a training program among COPD patients, without obtaining significant variations in the 6MWD [34]. Additionally, a systematic review [36] of 11 studies and 408 COPD patients reported the same conclusions and results, stating that the use of high-flow oxygen during a single session induced an increase in the functional exercise capacity (SMD 0.36, 95% CI 0.03–0.69, *p*-value 0.03), while the effect did not change in the overall analysis (*p*-value = 0.006).

We did not find an improvement in the 6MWD, but we did observe a change in the Borg scale for the perception of dyspnea and fatigue. This variation was statistically significant in both groups, and mostly in the experimental group, in line with studies by other authors [37]. In addition, we noted an improvement in the mMRC scale in the HFOT group. Previous studies did not identify changes in the Borg dyspnea and fatigue results between patients treated with HFOT and patients treated with oxygen, though a difference in the 6MWD was noted [34]. Schroff et al. observed that patients with the worst dyspnea and exercise capacity scores reported improvements greater than the minimum clinically significant difference. The authors concluded that patients with COPD showed significant improvements in the QoL, dyspnea score and exercise capacity, regardless of their baseline respiratory functional status, dyspnea score and exercise capacity [38]. In our opinion, the result of the *p* value in regard to the 6MWT does not have to be considered as negative. Indeed, the Borg dyspnea scale and exertional perception of the patients improved. Most patients in both groups described the training protocol as comfortable, and there was no significant difference in satisfaction between the two groups of patients. In the EG, the humidification delivered through HFOT may have played a positive role in ensuring comfort, and thus patients described the training as more comfortable.

Finally, we did observe any deaths or hospital readmissions of the included patients, and our qualitative analysis reported a trend of reduction regarding the exacerbation rates in both groups, dropping from a mean value of 1.47 ± 1.18 in the experimental group and of 1.68 ± 1.53 in the control group at T1 to a mean value of 0.18 ± 0.40 and 0.07 ± 0.25 in the experimental and control group, respectively, at the last follow-up.

Ranieri et al. reported a mortality rate of 20% at the 6-year follow-up in a group of older COPD patients discharged after a non-acidotic exacerbation [39].

Some studies have shown that the female gender is more predisposed to developing COPD, with a predominance of small airway disease, probably due to a sex-related differences in the expression and activity of cytochrome P450 enzymes. Moreover, women with severe COPD have a higher risk of hospitalization and death from respiratory failure [40]. In our study, we did not observe such differences, possibly because of the complexity and clinical severity of the patients included. However, this issue has not been investigated, and it will be the topic of future studies.

Murphy et al. found a 12-month risk of readmission or death equating to 63.4% in the home oxygen and home NIV treatment group vs. 80.4% in the home oxygen alone group, and 14% of the patients died in the home oxygen and home NIV group vs. 16% in the home oxygen alone group [7].

Gudmundsson et al., at a 24-month follow-up after hospital discharge following an exacerbation, found that 122 (29.3%) of the 416 patients had died [41]. The differences between these studies are probably due to the small number of enrolled patients and their similar levels of severity and treatment with home NIV.

### 4.1. Strengths of the Study

In our opinion, ours was the first study reported in the literature that enrolled COPD patients with chronic hypercapnic respiratory failure treated with LTOT and nocturnal NIV. It was performed on a selected and extremely severe cohort of COPD patients, among whom pharmacological treatment alone produces poor results.

We did not observe patient drop out, and this enabled a level of homogeneity in the evaluation of the data.

### 4.2. Limitations of the Study

Some limitations affected our study. Firstly, the small number of participants prevents us from drawing definitive conclusions, and further studies could provide important information that clinicians could use to quantify exercise training. Another limitation was the period of realization of this trial, which coincided with a lockdown in our country (Italy) of almost four months, which may have influenced our results. We recruited six patients in January 2020, but due to lockdown, we stopped and restarted the inclusion in September 2020. For our evaluation, we followed specific procedures. We performed spirometry only after the PCR test indicated the absence of SARS-CoV-2 in the patients in a room with an open window, using a UV system for sanitization. We performed a rapid antigen test before the training sessions, waiting for negative results before starting the training, and no positive case was identified.

## 5. Conclusions

In our study, we found that the sample of patients recruited showed improvements in their functional capacity, even though they were not statistically significant, and in their respiratory symptoms. This can be attributed to the application of a respiratory rehabilitation intervention. The use of HFOT during the training program did not result in variations between baseline and follow-up, although it improved the perception of dyspnea and fatigue and dyspnea scores. Although we observed no exacerbations or admissions, this cannot be related to the HFOT treatment.

## Figures and Tables

**Table 1 healthcare-10-02001-t001:** Measurements at each assessment time: mMRC (modified Medical Research Council scale); CAT (COPD assessment test); SGRQ (Saint George Respiratory Questionnaire); 6MWT (six-minute walking distance). T0: baseline; T1: at the end of the first 20 sessions of the rehabilitation cycle; T2: at the end of the second 20 sessions of the rehabilitation cycle; T3: at the end of the third 20 sessions of the rehabilitation cycle.

Time	Blood Gas Analysis	Spirometry	mMRC	6MWT	CAT	SGRQ	Borg Dyspnea
T0, baseline	X	X	X	X	X	X	X
T1	X		X	X	X		X
T2	X		X	X	X		X
T3	X	X	X	X	X	X	X

X indicates measurements evaluated at each assessment time.

**Table 2 healthcare-10-02001-t002:** Characteristics of patients at baseline. Results are shown as mean ± SD.

Characteristics	Experimental Group	Control Group	*p* Value
N (%)	15 (48.3)	16 (51.7)	
Age	75.7 (±7.50)	75.4 (±9.36)	
Female	60% (±0.50)	25% (±0.44)	<0.001
Active Smokers (n)	13% (2)	7% (1)	<0.001
p/y	46.1 (±33.9)	53.9 (±21.6)	
pH	7.41 (±0.016)	7.40 (±0.03)	0.579
PaCO_2_	53 (±5.95)	50.75 (±3.43)	0.237
PaO_2_	75.5 (±9.11)	67.9 (±12.5)	0.064
HCO^3-^	30.2 (±4.47)	27.7 (±4.60)	0.077
BE	5.96 (±5.14)	3.73 (±4.56)	0.122
FVC %	58.3 (±16)	58.3 (±19.8)	0.991
FEV1%	33.6 (±9.90)	44.19 (±13.6)	0.985
FEV1/FVC	44.5 (±10.4)	47.1 (±10.2)	0.774
DLCO	40 (±10.2)	44.4 (±16.9)	0.797
6MWD	222 (±68.2)	251 (±83.3)	0.902
Borg D PRE	1.93 (±2.1)	1.31 (±1.35)	0.340
Borg D POST	6.11 (±2.8)	5.8 (±2.73)	0.321
Borg F PRE	0.50 (±1.11)	1.03 (±1.11)	0.951
Borg F POST	3.34 (±3.55)	4.06 (±2.26)	0.834
CAT	19.93 (±5.66)	17.50 (±6.95)	0.095
mMRC	3.20 (±1.01)	3.19 (±0.91)	0.432
SGRQ	89.73 (±5.84)	90.44 (±6.45)	0.624

paCO_2_: arterial CO_2_ pressure; PaO_2_: arterial O_2_ pressure; HCO^3−^: blood bicarbonates; FVC: forced vital capacity; FEV1: forced expiratory volume in the 1st second; DLCO: CO lung diffusion; Borg D: Borg scale for dyspnea; Borg F: Borg scale for muscle fatigue; CAT: COPD assessment test; mMRC; modified Medical Research Council; SGRQ: Saint George Respiratory Questionnaire.

**Table 3 healthcare-10-02001-t003:** Borg scale variations between baseline (T0) and follow-up periods (T1,T2,T3. Borg scale results reported for the groups as the mean difference (±SD): mean difference and corresponding *p*-value. D1: dyspnea pre-6MWT; D2: dyspnea post-6MWT; F1: fatigue perception pre-6MWT; F2: fatigue perception post-6MWT.

Borg	Experimental	Control			Mean Difference	*p*-Value
	T0					
D1	1.93 (±2.1)	1.31 (±1.35)	T1			
			1.92 (2.10)	1.31 (1.35)	0.01	1.00
			T2			
			2 (2.12)	1.12 (1.0)	1.50	0.047
			T3			
			1.07 (±1.04)	1.00 (±0.63)	1.00	0.01
D2	6.11 (±2.8)	5.8 (±2.73)	T1			
			4.80 (1.95)	4.75 (1.69)	2.00	<0.001
			T2			
			5.11 (1.36)	4.44 (1.26)	2.01	<0.001
			T3			
			4.0 (±1.6)	3.68 (±1.3)	2.50	<0.001
F1	0.50 (±1.11)	1.03 (±1.11)	T1			
			0.5 (1.12)	1.03 (1.12)	0.001	1.00
			T2			
			0.61 (1.32)	1.03 (1.12)	0.01	1.00
			T3			
			0.31 (±0.63)	0.50 (±0.63)	1.00	0.003
F2	3.34 (±3.55)	4.06 (±2.26)	T1			
			2.92 (2.6)	3.44 (1.63)	1.00	0.005
			T2			
			3.11 (2.98)	3.44 (1.63)	1.00	0.005
			T3			
			2.1 (±2.1)	2.63 (±1.3)	2.00	<0.001

## Data Availability

If you need data, please ask from correspondence.

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
