# Peer review of "Efficacy of Nasal High-Flow Oxygen Therapy in Chronic Obstructive Pulmonary Disease Patients in Long-Term Oxygen and Nocturnal Non-Invasive Ventilation during Exercise Training"

_healthcare, 2022, doi:10.3390/healthcare10102001_

Round 1
Reviewer 1 Report
Please explain the study strengths in the section with limitations, and comment on the influence of COVID-19 pandemics on the study results, since the period of the survey is within the pandemics, lockdowns, difficulties in performing some respiratory functional diagnostics (spirometry, etc.).
Reviewer 2 Report
This manuscript investigates the effects and efficacy of pulmonary rehabilitation with the addition of HFOT in COPD patients in nocturnal Non-Invasive Ventilation and long-term oxygen therapy (LTOT). The idea is interesting and important for the literature. However, I have some comments that the authors might want to take into consideration to improve clarity of the manuscript. See below:
1- As there was a statistical difference between the first two follow-ups (T0-T1 and T0-T2), these data should also be presented in the table, as was done for the last follow-up (Table 3).
2- The authors showed that statistical significance was found only for the BORG scale regarding dyspnea but there is no emphasis on the importance of improving the score on this scale. The authors should discuss the importance of the Borg scale and the impact on the quality of life of patients by improving this score.
3- Please discuss the mechanism that would explain why improvement on the BORG scale can occur in the absence of improvement in functional parameters.
Minor:
1- Abstract (line 12): The sentence: "To evaluate the efficacy..." is confusing. I believe the authors meant that: “This study aimed to evaluate…”
2- Introduction (line 37): "Murphy in his study shows" it should be: "Murphy et al., 2017, showed…"
The same in Discussion (line 258): "In Murphy et al. study it was found…", should be: "In the study by Murphy et al., 2017, it was found…"
3- The authors do not report the method of withdrawal of arterial blood (which artery, the volume of blood withdrawn...)
4- Limitations of study (line 318): The authors state that the period of realization of the trial coincided with a four months period of lockdown in their country and this may have influenced the results. It was not clear whether during this lockdown period the protocols had to be interrupted. If yes, this information must be included in the protocol description.
5- Figure S3 (line 238): The legend should be "Figure S3" instead of "Figure 3".
6- Table 3 is listed as supplementary material but I think it was a mistake because it seems to be a regular, non-supplementary table.
Reviewer 3 Report
The authors sought to investigate the efficacy of high flow nasal therapy in chronic obstructive pulmonary disease (COPD) patients with long-term oxygen therapy (LTOT) and nocturnal non-invasive ventilation (NIV). Whereas the results are clearly presented and the design sufficiently described, there are several points that need to be further addressed:
1) First two sentences of the conclusion relate to the data that are not presented in your manuscript. Please add the table with this data (or cite it if you already published it elsewhere).
2) In the discussion (lines 290-300), you are mentioning gender differences in susceptibility for the development of different clinical phenotypes of COPD. Although I don't find this topic particularly relevant for your current study, if you decide to keep it in your revised manuscript, please provide a more thorough overview of the available literature. It has been suggested that observed differences arise from exposure to different risk factors (biomass smoke versus tobacco smoke), or even from different smoking patterns between genders. Moreover, as you did not have any death or hospital readmission of included patients, it is unclear how the lack of differences in these aspects can be explained by "the complexity and clinical severity of patients included". Please rephrase.
3) First sentence of the introduction: although this is usually the case, COPD patients do not always have frequent periods of exacerbations. please rephrase the sentence accordingly.
4) Please revise the abstract to make it unstructured with grammatically coherent, full sentences. Please check the rest of the manuscript as well: verbs are often missing from the sentences and the use of present and past tense is not consistent.
